# Using a low-dose ultraviolet-B lighting solution during working hours: An explorative investigation towards the effectivity in maintaining healthy vitamin D levels

**Laura M. Huiberts**[1], **Karin C. H. J. Smolders**[1], **Bianca M. I. van der Zande**[2], **Rémy C. Broersma**[2], **Yvonne A. W. de Kort**[1] *

**1** Human-Technology Interaction, School of Innovation Sciences, Eindhoven University of Technology, Eindhoven, the Netherlands, **2** Signify Research, Eindhoven, the Netherlands

* y.a.w.d.kort@tue.nl

## Abstract

### Objective

This study examined whether daily safe, low-dose ultraviolet-B (UVB) exposure using a home-based lighting solution could maintain healthy serum 25(OH)D during winter.

### Methods

Twenty-eight (12 male, 16 female) daytime (~9:00 to 17:00) indoor workers (mean age = 42.46; $SD$ = 14.23) participated in this study and were allocated to one of the two study conditions. During an 8-week period, fourteen participants received extra UVB exposure (max 0.3 standard erythema dose (SED) daily), while fourteen participants in the control group did not receive extra UVB exposure. Daily questionnaires were used to measure UVB exposure time, exposed body surface area (BSA), and time spent outside in daylight. Serum 25(OH)D, vitamin D related food intake, and secondary parameters (i.e., subjective fatigue, sleep timing and quality) were investigated at baseline, Week 4, and Week 8.

### Results

Serum 25(OH)D significantly declined over the 8-week study period in both groups. The combination of using a low-dose UVB exposure, a small BSA, and a lower-than-expected amount of exposure hours likely resulted in an insufficient UVB dose to significantly improve serum 25(OH)D. Changes in serum 25(OH)D over time did not significantly correlate with changes in secondary parameters of sleep and fatigue.

### Conclusion

The received low-dose UVB exposure in this study did not significantly change serum 25(OH)D during the winter period. Future research could explore whether a longer lasting exposure period and/or using different exposure positions of the device (maximizing exposed skin surface) yields more promising results for improving serum 25(OH)D.

**Data Availability Statement:** Data cannot be shared publicly because of the informed consent

form stating that data will be only available to others in an encrypted and password protected institutional online data repository. Data are available upon request via the OSF project page through which requests can be sent to the authors (https://osf.io/qnk26/) or via the institutional Ethics Committee (contact via ethicalreviewboardHTI@tue.nl).

**Funding:** This research was funded by Signify, Eindhoven, the Netherlands. Funders had a role in the study design, data collection (only light measurements), and preparation of the manuscript. Funders had no role in data collection from participants, data analysis, and decision to publish.

**Competing interests:** The authors have declared that no competing interests exist.

## Trial registration

Trial registration: https://www.isrctn.com/ISRCTN47902923.

## 1. Introduction

Vitamin D insufficiency ($< 30$ ng/ml; $<75$ nmol/L) affects more than 70% of the general population worldwide [1] and has been associated with health consequences such as osteoporosis, cancer, and autoimmune disorders in the long term [2,3], and increased susceptibility to viral and bacterial infections in the short term [4]. The main route of vitamin D synthesis is via ultraviolet-B (UVB) exposure. However, due to indoor lifestyles (~70–90% of our time [5–7]) and most of the sunlight's UVB radiation being absorbed by the ozone layer at higher latitudes [8], many people miss out on the benefits of UVB exposure, even in summer season [9]. Moreover, food sources alone do not contain adequate vitamin D concentrations to ensure sufficient ($> 75$ nmol/L) serum vitamin D (25(OH)D) levels [10,11].

Recent studies also revealed a potentially important role of daily vitamin D synthesis in facilitating mental wellbeing and sleep [12], due to its role in serotonin and melatonin regulation [13,14]. Given the high prevalence of vitamin D insufficiency, increasing Vitamin D status in the general population is expected to positively influence feelings of wellbeing and sleep quality [12]. Therefore, ensuring sufficient vitamin D levels may not only decrease health care costs but also economic costs by increasing productivity at work and preventing sick leave [15].

Daily supplementation with vitamin D can be a good solution to reach sufficient serum 25 (OH)D. However, adherence to a vitamin D supplementation scheme is very important to sustain sufficient levels, which may not be easy for everyone. Moreover, vitamin D synthesis via UVB exposure is expected to be more efficient in raising serum 25(OH)D compared to supplement intake [16,17]. Last, UVB exposure also exclusively leads to the synthesis of different vitamin D metabolites in the skin which all have unique contributions to health and wellbeing, such as anti-inflammatory effects [18].

Because of the above-mentioned benefits, receiving low-dose artificial UVB exposure in winter seems an unobtrusive solution to improve serum 25(OH)D. Previous controlled and uncontrolled studies using UV light sources to improve serum 25(OH)D revealed that exposure to relatively low doses of UVB may be adequate for maintaining healthy 25(OH)D levels throughout winter [17,19–25]. For example, a previous dose-response study showed that a cumulative standard erythema dose (SED) of 8 delivered over a 16-week period already proved to be sufficient to sustain summer 25(OH)D levels in winter compared to receiving no extra UVB exposure [24]. However, these previous studies generally used sunbed applications or (medical) treatment lamps (e.g., for psoriasis), exposed relatively large body surface areas (BSA's), and used exposure protocols that interfere with people's daily routines as they require visits to a hospital or clinic.

To date, no studies have been reported that investigated the effectiveness of prolonged daily low-dose UVB exposure in the home environment using a relatively small BSA on serum 25 (OH)D. Therefore, we explored for the first time whether a low and safe [26] dose of UVB exposure applied in a home office lighting solution would be sufficient to maintain healthy serum 25(OH)D during winter in daytime (~9:00 to 17:00) indoor workers wearing their regular clothing. Taking into account the relatively small BSA in our study (generally hands, face and neck), we hypothesized that a cumulative dose of 12 SED (i.e., a maximum daily dose of 0.3 SED) would be sufficient to achieve an attenuated decline in serum 25(OH)D during the

8-week winter period. In the control group, a sharp decline in serum 25(OH)D was expected as is usually the case during the winter period in people not taking supplements [27,28]. Last, as a secondary aim we explored correlations between changes in serum 25(OH)D and sleep indicators (midsleep, sleep duration and quality) and general levels of daytime fatigue over time.

## 2. Materials & methods

### 2.1. Design

A between-subjects design was employed to investigate the effect of low-dose vs. no additional UVB exposure during working hours on serum 25(OH)D. Randomization to the study conditions was realized where possible (see section *2.2.*). UVB vs. no UVB exposure was the independent variable and serum 25(OH)D was the primary dependent variable in the study. The study was approved by the medical ethical committee of the Maxima Medical Center Veldhoven, The Netherlands. Trial registration on ISRCTN (ISRCTN47902923) was carried out retrospectively so that the developed technology did not have to be shared publicly at an early stage. The full study protocol is also available from ISRCTN. Authors confirm that all ongoing and related trials for this intervention are registered.

### 2.2. Participants

All participants in this study were recruited and checked for participation eligibility by an independent recruitment organization to ensure independency of the research team. Recruitment was conducted from November 23rd through December 18th 2020. Inclusion and exclusion criteria for participation can be viewed in Table 1.

A conservative sample size calculation based on the serum 25(OH)D simulation model proposed by Diffey [30] was conducted in G*Power before the start of the recruitment phase. For our primary research question (i.e., the effectiveness of daily low-dose UVB exposure vs. no extra UVB exposure on serum 25(OH)D) the sample size calculation using an expected exposure of at least 0.225 SED during workdays, a BSA exposure of 10.5%, and a power of 0.90 required a total sample size of 34 (17 in each group). The extensive description of the sample size calculation can be found in the study protocol file (available from ISRCTN47902923). Due

**Table 1. In- and exclusion criteria for participation.**

| Inclusion criteria | Exclusion criteria |
|---|---|
| • Age 18 or older | • Current pregnancy, breast feeding or a desire to become pregnant |
| • Fitzpatrick skin type II or III | • Having children at home aged 10 years or younger |
| • Living in/around Eindhoven | • Having malignant skin conditions in the past or currently |
| • Medically fit to work the hours as contractually agreed | • Photosensitive medical conditions or photo-sensitising drugs |
| • Desk presence (at the home office) of at least 2.5 days per week during the 8 weeks study | • Users of medicines and/or crèmes mentioning in the prescription as side effect extra sensitivity to the sun / interaction with sun exposure |
| • Finding it no problem to have blood drawn three times over a period of 8 weeks | • Planned use of sun beds, or sunbed use during the past 4 weeks |
| | • Currently taking or planning to take oral vitamin D supplements or have been taking D supplements during the past 4 weeks |
| | • High vitamin D levels at the start of the study (>375 nmol/L) which need medical attention [29] |

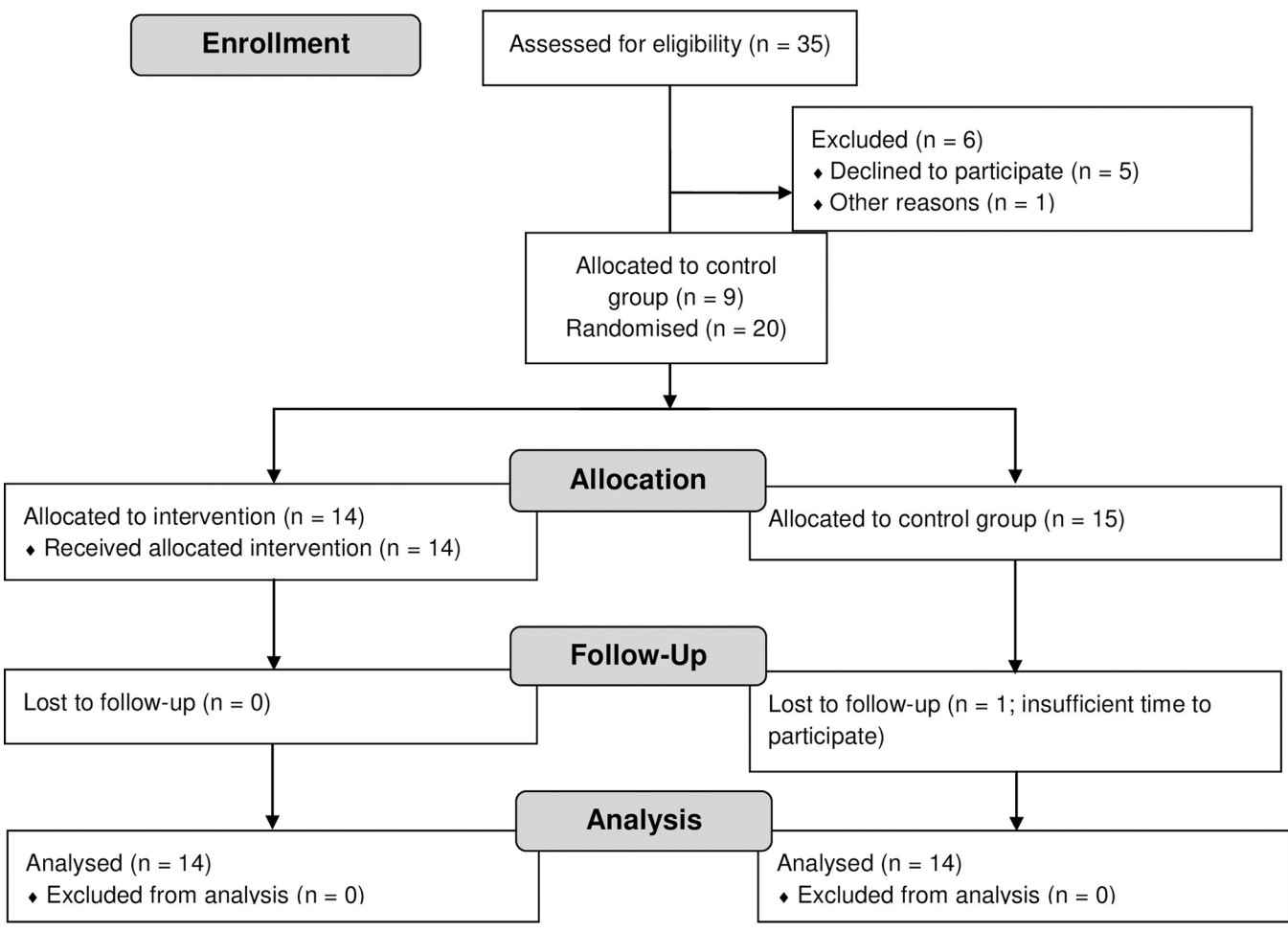

**Fig 1. Participant allocation flowchart.**

to the availability of 14 devices for this study, a total sample size of at least 28 participants (14 in each group) was aimed for during the recruitment phase.

At the end of the recruitment phase, 35 eligible participants were recruited and informed about the study. From these, 29 signed the informed consent form for participation in the study. Due to the predetermined study period timing (January to March for all participants), it was decided to start the study with 29 participants who gave written consent. Participants were randomly allocated to the two study conditions via randomization as much as possible (see Fig 1). However, this was not possible for several participants as they either changed in their work location or number of working hours at home before the start of the study ($N = 5$), because they did not have grounded electricity outlets in their houses needed for the installation of the lamp ($N = 3$), or because there was insufficient space for installation of the lamp ($N = 1$). This information was obtained after the recruitment phase had ended, and these participants were still willing to participate in the study. To this end, nine participants were allocated to the control group before randomization was carried out for the remainder of the participants. Subsequently, randomization was conducted so that 14 participants were allocated to the intervention group, and six participants were allocated to the control group. This was realized by generating a random sequence of six zeroes (control group) and 14 ones (intervention group) and adding participants to this sequence based on date of returning the signed

informed consent form. Randomization was conducted by the principal investigator (first author) of the study. One participant allocated to the control group dropped out at the beginning of the study due to job changes which resulted in too little time to participate. In addition, although vitamin D supplement use was an exclusion criterion, one participant in the control group indicated in the second week of the study to take vitamin D in a daily multivitamin (10 micrograms daily). It was decided to leave this participant in the study by asking her to continue to take this supplement daily to keep conditions constant over the study period. The main analyses were conducted with and without this participant.

## 2.3. Setting and materials

The study was performed in participants' home offices as COVID-19 regulations asked people to work from home during the study period (January 2021 to March 2021 in The Netherlands). For participants in the intervention group, a lamp was installed at the home office before the start of the study. In the visual wavelength range, the lamp provided an illuminance of about 1100 lx (~ 3700 K) at 20 cm distance and about 100 lx at 80 cm distance. Additionally, the lamp emitted a continuous ultra-low and safe artificial actinic UV irradiance of less than 3.0 $mW_A/m^2$ at 20 cm distance and 0.77 $mW_A/m^2$ at 80 cm distance [26]. This resulted in an estimated maximal received dose of ~0.3 SED (UVI = 0.04) per 8-hour workday when seated at the desk. Participants in the intervention group and members of their household were advised to respect the indicated minimum application distance (80 cm). This distance keeps the actinic UV exposure within Risk Group Exempt limits, which allows for a continuous exposure of 30000 sec per day (8 hrs. and 20 minutes) according to the IEC62471 [26]. The UVB irradiance stability was checked by a before and after actinic UVB measurement at the Lighting Test Center Europe, Eindhoven. The equipment used for these measurements consisted of a Cary 17d monochromator and a Keithley 2000 multimeter. The scanned spectral range was 200–800 nm and measurements were executed at 80 cm distance. The average actinic UV weighted UVB irradiance ($mW_A/m^2$) of the 14 lamps before and after the study were respectively 0.76 $mW_A/m^2$ (with a slight variation across lamps: SD = 0.04 $mW_A/m^2$) and 0.75 $mW_A/m^2$, (SD = 0.07 $mW_A/m^2$). UV Irradiance stability during the study period was checked by measuring UV-intensity in Week 1, Week 4, and Week 8 using the Shade UVR sensor at 80 and 20 cm distance from the lamp. This deviated from the original study protocol, in which weekly solar meter measurements and continuous UV measurements at the table were proposed. Three instead of weekly UV measurements were chosen to minimize the number of visits to participants' homes, which was strongly advised during this time of the COVID-19 pandemic in the Netherlands. In addition, only the Shade UVR sensor was used for measurements as it is much more sensitive than the solar meter. Continuous measurement at the home office table was discarded due to potential privacy issues with the Shade device, which had to be connected to participants' personal smartphone while logging personal information. The irradiance stability throughout the study period was realized by an adjustment of the Digital Addressable Lighting Interface (DALI) settings at the end of Week 4 to correct for the slow decline in UV intensity due to the lifetime degradation of the used UVB LEDs. Participants in the control group did not receive a lamp in their home office and were therefore not exposed to extra UVB.

## 2.4. Measurements

**2.4.1. Serum vitamin D.** Blood samples were taken three times during the study period (Baseline, Week 4, and Week 8) to establish serum 25(OH)D in nmol/L. Samples were analyzed by an independent local medical laboratory (Diagnostiek Voor U) using the automated

immunoassay method. Due to potential fluctuations in serum 25(OH)D over the day [31,32], participants were asked to have their blood sampled at approximately the same time of day.

**2.4.2. Sleep, fatigue, and vitamin D related food intake.** Sleep timing and quality, fatigue, and vitamin D related food intake were measured three times during the study period (Baseline, Week 4, and Week 8) using an online questionnaire. Sleep timing and quality were measured by integrating items from the Consensus sleep diary [33] and the Munich Chronotype Questionnaire [34], adapted for measurement over the past two weeks. Sleep duration was calculated by the difference in time between sleep onset and sleep offset, minus the total amount of minutes lying awake at night. Midsleep timing was calculated by adding half of the total sleep duration to the sleep onset time. Sleep quality was the average score of four items measured on a 5-point Likert scale. These four items probed 1) overall sleep quality; 2) easiness to fall asleep; 3) feeling refreshed at wake-up; and 4) easiness to get out of bed (Cronbach's α = 0.79). Last, subjective fatigue was measured using the Checklist Individual Strength (CIS) [35], the sum score was used as outcome variable. Vitamin D related food intake during the past four weeks was measured using a short questionnaire previously developed by Hedlund et al. [36], including intake of (fortified) cereals.

**2.4.3. Daily diaries.** Daily diaries at the end of each workday measured time spent working at the desk (exposure duration); time spent outside (on Mondays reporting of time spent outside in the weekend was added); and exposed skin surface based on clothing (intervention group only). Participants received a daily link to this online questionnaire via e-mail between 17:00 and 18:00.

## 2.5. Procedure

One month before the start of the study, participants were called by the responsible researcher to receive information about the procedure of the study and double-check the in- and exclusion criteria. Afterwards, participants were sent the study information and consent form, and returned the signed consent form to the researcher via mail when they decided to participate. During the 1.5-week period before the start of the study, participants completed the baseline questionnaire and the first blood sampling measurement. In addition, the lamps were installed in the intervention group's homes by a technician. They received additional instructions about the lamp functionalities and behavioral guidelines during the study period. Subsequently, the 8-week study period started for both groups (January 8, 2021). Monday-Fridays, the lamps turned on automatically at 8:50 and off at 17:10. In weekends, and between 17:11 and 8:49 the lamp was switched off (i.e., no visual light and no UVB exposure). Participants received the daily questionnaire at the end of each workday. In Week 4 and 8, the blood sampling and the main questionnaire were repeated for all participants. After Week 8, the lamps were deinstalled by the technician and participants received a financial compensation for their participation. Participants received a debriefing and overview of the main findings one month after the end of the study.

## 2.6. Statistical analyses

All statistics were performed in IBM SPSS Statistics 23.0. Baseline measurements were checked for potential outliers (more than three standard deviations from the mean value). In case any outliers were found, analyses were run with and without the outlier value(s).

First, descriptive statistics of the baseline variables for both conditions were reported and compared between the two groups via independent-samples *t* tests. Subsequently, to investigate the main research aim (i.e., effects of the UVB manipulation on serum 25(OH)D) an ANCOVA analysis with condition as predictor and change scores in serum 25(OH)D (baseline

to Week 8) as outcome variable was conducted. This analysis was controlled for cumulative time spent outside (centered). The assumption for homogeneity of variances was met according to the Levene's test (F = 0.20; $p$ = 0.66). In addition, homogeneity of regression slopes was met as there was no interaction between time spent outside and condition (F = 0.46; $p$ = 0.64)

To further investigate potential effects of the intervention over time (i.e., effects of the UVB manipulation on serum 25(OH)D at Week 4 and Week 8), alinear mixed model (LMM) analysis with participant ID as a random intercept, controlling for cumulative time spent outside (centered) and baseline serum 25(OH)D (centered) was conducted. In case a significant condition * measurement (time) interaction was found ($p$ < 0.05), post-hoc comparisons were conducted to investigate differences between groups at the last measurement point (Week 8), and at Week 4 to explore intermediate effects.

For the intervention group only, we conducted exploratory Pearson correlation analyses investigating the association between cumulative desk presence (UVB exposure) until Week 4 and until Week 8 and respectively change in serum 25(OH)D from baseline to Week 4 and Week 8. The same was analyzed for the control group. However, in case of the control group cumulative desk time did not reflect any additional UVB exposure.

Last, exploratory Pearson correlations were conducted to investigate the secondary study aims, namely the relationships between changes in serum 25(OH)D and changes in sleep duration, midsleep timing, sleep quality, and general fatigue over time across all participants (Baseline to Week 8). For visual inspection purposes only, plots were created regarding associations between serum 25(OH)D and each of the subjective outcome variables. Corresponding statistics from LMM analyses, including intervention and measurement week as fixed factors, were reported with these plots in the Supportive Information.

For each of the LMM analyses, symmetry, independence, and homoscedasticity of the residuals were visually inspected by plotting the residuals against the fitted predicted values of each model. In addition, homoscedasticity was further confirmed by investigating Pearson correlations between the absolute residuals and the standardized predicted values (which should be low and non-significant). In addition, normality of the residuals were evaluated by conducting Shapiro-Wilk tests. For the main LMM analysis (section 3.3.) residuals were normally distributed (Shapiro-Wilk test: W = 0.98; $p$ = 0.37), and there was a low non-significant correlation between the standardized predicted values and the residuals ($r$ = 0.20; $p$ = 0.14). For the exploratory LMM analyses (section 3.5) residuals were normally distributed for all models (Shapiro-Wilk tests: all $p$'s > 0.20) except for the model investigating associations between serum 25(OH)D and sleep duration in the intervention group (Shapiro-Wilk test: W = 0.93; $p$ = 0.03). For most models, there was a low non-significant correlation between the standardized predicted values and the residuals (all $r$'s < 0.32; all $p$'s > 0.05). Only for the model investigating associations between serum 25(OH)D and midsleep in the intervention group the correlation just reached significance ($r$ = 0.33; $p$ = 0.049). However, as we only conducted these analyses for visual inspection purposes we decided to pursue these analyses.

## 3. Results

### 3.1. Response rates and descriptive statistics at baseline

Table 2 shows the demographic data of all participants and per study condition. There was no significant difference in age between groups ($t$(26) = -0.30; $p$ = 0.77), neither between gender distribution across groups ($\chi^2$ = 0.58; $p$ = 0.45).

All three assessments (i.e., blood samplings and main questionnaires at baseline, Week 4, and Week 8) were 100% completed. Only two participants in the intervention group missed some of the daily questionnaires (ten of 560 daily questionnaires were missing), while three

**Table 2. Demographic data of the total study sample and each group.**

|  | Total sample | Intervention group | Control group |
|---|---|---|---|
| *N* | 28 | 14 | 14 |
| Gender |  |  |  |
| Male (*N*) | 12 | 7 | 5 |
| Female (*N*) | 16 | 7 | 9 |
| Age (M, SD) | 42.45 (14.23) | 43.29 (16.26) | 41.64 (12.44) |

participants in the control group missed some daily questionnaires (eight of 560 daily questionnaires were missing). In addition, three observations in the sleep duration and midsleep variables as assessed in the baseline questionnaire were coded as missing as the reported sleep-timing variables were uninterpretable due to unclear reporting of the participant. No outliers (values > 3 SD's from the mean) were found for variables that were used in the analyses for hypothesis testing. Table 3 shows the descriptive statistics of the variables measured at baseline. Independent-samples *t* tests at baseline revealed that only sleep duration on workdays significantly differed between the two groups ($t(23) = 2.52$; $p = 0.02$).

## 3.2. Descriptive statistics daily questionnaires

On average, participants in the intervention group were exposed to the lamp for 23.67 hours per week (*SD* = 5.39; Min = 15.59; Max = 31.83). Table 4 shows the cumulative amount of exposure hours until Week 4 and Week 8. In addition, Table 4 shows the cumulative amount

**Table 3. Descriptive statistics of all variables measured at baseline.**

|  | Intervention group (*N* = 14) | | | | Control group (*N* = 14) | | | |
|---|---|---|---|---|---|---|---|---|
|  | M | SD | Min | Max | M | SD | Min | Max |
| Serum 25(OH)D (nmol/L) | 59.00 | 15.36 | 42 | 97 | 57.79 | 19.53 | 28 | 84 |
| Fish intake[1] (grams) | 365.00 | 367.96 | 0 | 1200 | 262.86 | 320.06 | 0 | 1200 |
| Milk intake[1] (litres) | 1.92 | 2.49 | 0 | 7.00 | 1.83 | 2.92 | 0 | 8.40 |
| Yoghurt intake[1] (litres) | 1.43 | 2.15 | 0 | 7.00 | 1.85 | 2.53 | 0 | 8.00 |
| Butter intake[1] (portions on bread) | 67.00 | 87.48 | 0 | 224 | 32.71 | 45.27 | 0 | 168 |
| Sleep duration (hours) | 6.54 (N = 12) | 0.75 | 5.25 | 7.67 | 7.59 (N = 13) | 1.25 | 5.25 | 9.75* |
| Midsleep (hh:mm) | 03:23 (N = 12) | 00:45 | 02:00 | 04:30 | 03:26 (N = 13) | 00:59 | 01:38 | 04:45 |
| Sleep quality (scale 1–5) | 3.16 | 0.86 | 1.00 | 4.50 | 3.36 | 0.71 | 1.75 | 4.50 |
| Time spent outside workdays (minutes per day) | 66.79 | 61.73 | 20 | 260 | 87.50 | 68.80 | 10 | 250 |
| Time spent outside free days (minutes per day) | 84.86 | 41.81 | 3 | 150 | 139.29 | 118.69 | 20 | 500 |
| Subjective fatigue; CIS total score[2] (range: 20–140) | 68.29 | 24.14 | 21 | 110 | 66.64 | 21.27 | 23 | 95 |

* $p < 0.05$.

[1] Total intake during the past four weeks.

[2] A higher score indicates more subjective fatigue.

*Note.* cereal consumption was not further reported here, as only one participant in the control group ate a cereal brand with a small amount of added vitamin D.

**Table 4. Cumulative exposure time and time spent outside in hours over the study period.**

| | Intervention group | | | | Control group | | | |
|---|---|---|---|---|---|---|---|---|
| | M | SD | Min | Max | M | SD | Min | Max |
| Lamp exposure Week 4 (midterm) | 92.30 | 18.86 | 51.25 | 124.50 | - | - | - | - |
| Lamp exposure Week 8 | 189.33 | 43.13 | 124.75 | 254.63 | - | - | - | - |
| Time spent outside Week 4 (midterm) | 20.93 | 8.46 | 8.25 | 39.25 | 29.05 | 20.44 | 5.69 | 87.70 |
| Time spent outside Week 8 | 50.06 | 15.47 | 28.42 | 81.25 | 66.08 | 39.06 | 22.86 | 179.20 |

of time spent outside in hours per group. There was no statistically significant difference between the two groups in Week 4 ($t = 1.37$; $p = 0.18$) or Week 8 ($t = 1.42$; $p = 0.17$) in the cumulative time spent outside. The Cohen's $d$ was moderate; respectively 0.52 and 0.54. Lastly, exposed BSA was inspected in the intervention group. The hands and face (BSA ≈ 9%) were exposed in most cases (resp. 96.8% and 96.7%), while the neck area (BSA ≈ 1%) was exposed in 81.1% of the cases. The upper area of the head (BSA ≈ 1%) was additionally exposed in two persons, who had little to no hair. Finally, three of the 14 participants had their (fore)arms (BSA ≈ 4%) exposed on most of the days during the study period, either by wearing a T-shirt or by rolling up their sleeves.

### 3.3. Effects of the intervention on serum 25(OH)D

Investigating our main aim, the ANCOVA findings revealed that the intervention did not significantly affect changes in serum 25(OH)D from baseline to Week 8 (controlled for time spent outside). The statistics of this analysis can be viewed in Table 5. The analysis was conducted with and without the participant in the control group taking a daily multivitamin containing vitamin D. As excluding the participant did not affect the interpretation of the results, it was decided to report the findings including all participants ($N = 28$) in the analysis. The findings of the LMM analysis investigating the effect of the intervention on serum 25(OH)D as a function of measurement week (Week 4 vs. 8) controlled for baseline serum 25(OH)D and cumulative time spent outside are shown in Table 6. This analysis was conducted with and without the participant in the control group taking a daily multivitamin containing vitamin D. As excluding the participant did not affect the interpretation of the results, it was decided to report the findings including all participants ($N = 28$) in the analysis. The interaction term in the model revealed no significantly assessment-dependent moderation in serum 25(OH)D between conditions (see Table 6). There was also no significant relationship between cumulative time spent outside during the past four weeks and serum 25(OH)D. The decrease in serum 25(OH)D from Week 4 to Week 8 also did not reach statistical significance. As

**Table 5. Statistics ANCOVA; effect of the intervention on change in serum 25(OHD) from baseline to Week 8.**

| | Estimate (unstandardized coefficient) | SE | 95% CI | t-value | p-value | $\eta_p^2$ |
|---|---|---|---|---|---|---|
| Intercept | -12.57 | 2.93 | -18.60 – -6.55 | -4.30 | <0.001 | 0.43 |
| Condition[1] | -0.567 | 4.22 | -9.25–8.12 | -0.14 | 0.89 | 0.001 |
| Time spent outside[2] | 0.001 | 0.001 | -0.001–0.003 | -0.88 | 0.39 | 0.03 |

[1] Condition is coded 0 for the control group and 1 for the intervention group.

[2] Specified as cumulative time spent outside in minutes during the past 4 weeks (centered).

**Table 6. Statistics linear mixed model analysis; effect of the intervention and measurement on serum 25(OH)D measured in Week 4 and Week 8.**

| Fixed effects | Estimate (unstandardized coefficient) | SE | 95% CI | t-value | p-value |
|---|---|---|---|---|---|
| Intercept | 47.22 | 2.06 | 43.06–51.37 | 22.91 | < .001 |
| Condition[1] | 3.30 | 2.99 | -2.73–9.33 | 1.10 | 0.57 |
| Measurement[2] | -1.94 | 2.09 | -6.20 – -2.32 | -0.93 | 0.36 |
| Baseline serum[3] 25(OH)D | 0.59 | 0.07 | 0.44–0.75 | 7.98 | < .001 |
| Time spent outside[3,4] | 0.002 | 0.001 | -0.00–0.00 | 1.19 | 0.25 |
| Condition * Measurement | -3.59 | 2.81 | -9.35–2.16 | -1.28 | 0.21 |
| **Random effects** | **Estimate** | **SE** | **95% CI** | **z** | **p-value** |
| Level 2 Intercept | 31.85 | 12.79 | 14.50–69.98 | 2.49 | 0.01 |
| Level 1 Residual | 27.60 | 7.39 | 16.33–46.66 | 3.73 | <0.001 |

[1] Condition is coded 0 for the control group and 1 for the intervention group.

[2] Measurement is coded 0 for Week 4 and 1 for Week 8 serum 25(OH)D measurements.

[3] Baseline serum 25(OH)D and Time spent outside were grand mean centered in this model.

[4] Specified as cumulative time spent outside in minutes during the past 4 weeks.

expected, there was a significant positive relationship between baseline serum 25(OH)D and subsequent measures of serum 25(OH)D. Fig 2 visualizes the change in serum 25(OH)D over time for both groups using the values calculated from this model. At Week 4, the intervention group had–on average–a higher serum 25(OH)D than the control group (mean difference = 3.30; SE = 2.99; 95% CI -2.73–9.33), but this difference did not reach statistical significance ($p = 0.28$). The difference between the groups at Week 8 was also non-significant (mean difference = -0.29; SE = 2.99; 95% CI -6.31–5.73; $p = 0.92$).

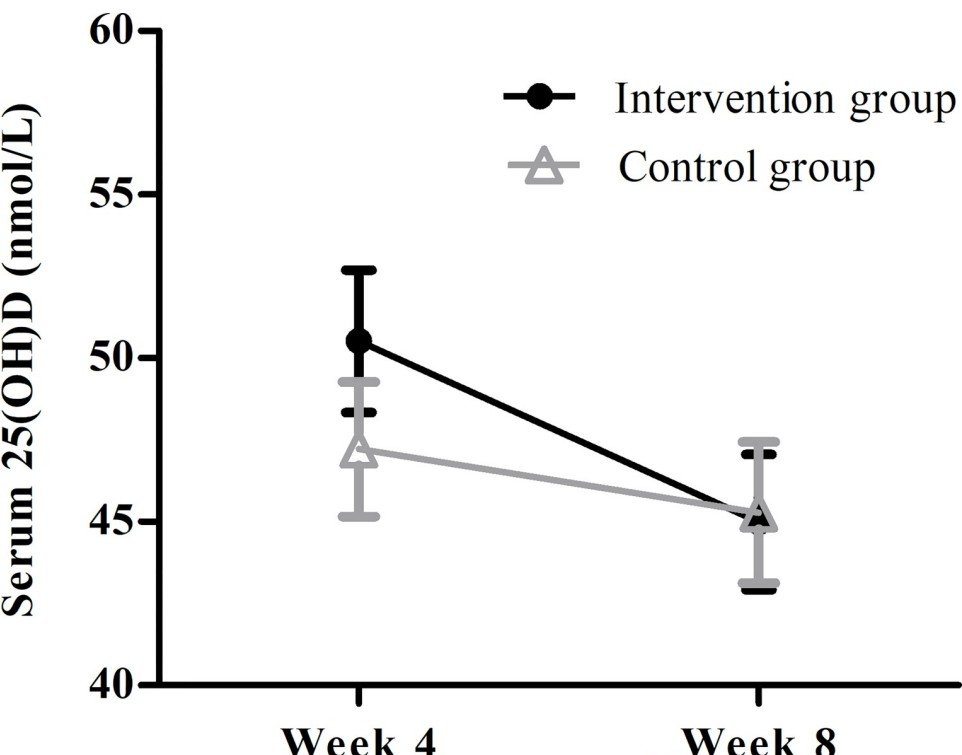

**Fig 2. Average serum 25(OH)D over time (Week 4 and 8) in both groups corrected for baseline values and time spent outside.** Error bars represent SE's. The study period was from January (Baseline) to March (Week 8).

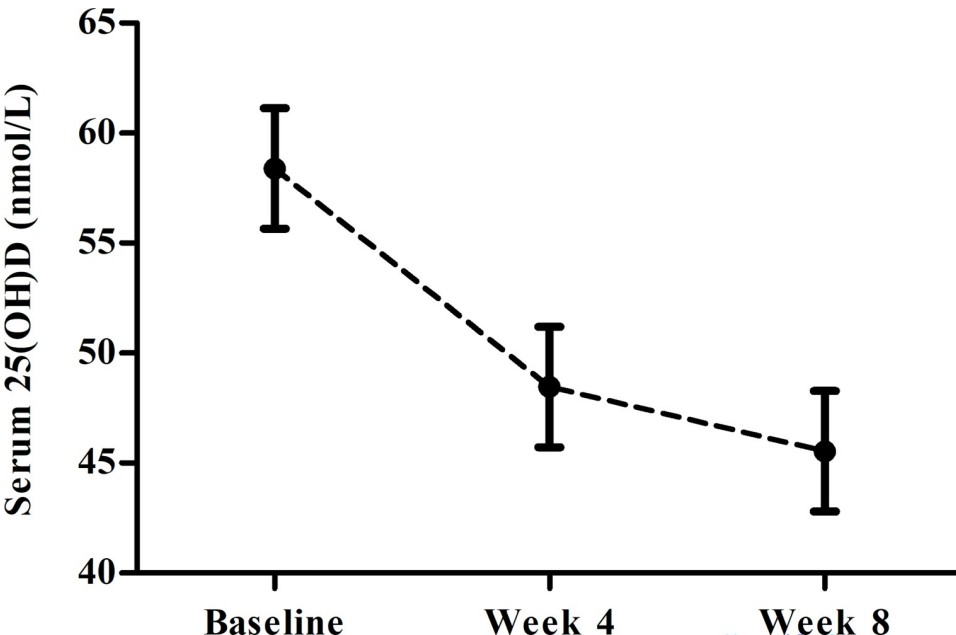

**Fig 3. Average serum 25(OH)D over time (Baseline to Week 8) across all participants, error bars represent *SE*'s.** The study period was from January (Baseline) to March (Week 8).

As there were no significant differences in serum 25(OH)D between the two groups, an additional LMM analysis with measurement as predictor (Week 1, 4, and 8) was conducted to visualize the decline in serum 25(OH)D over time across all participants. This analysis revealed an overall highly significant effect of Measurement ($F(1, 45)$ = 26.04; $p < 0.001$). Post hoc tests revealed a significant decline in serum 25(OH)D from Week 1 to Week 8 (mean difference = -12.86; $SE$ = 1.87; $t$ = -6.87; $p < 0.001$) and from Week 1 to Week 4 (mean difference = -9.93; $SE$ = 1.85; $t$ = -5.37; $p < 0.001$), but not from Week 4 to Week 8 (mean difference = -2.93; $SE$ = 1.85; $t$ = -1.59; $p = 0.12$). Fig 3 visualizes the decline in serum 25(OH)D across all participants.

### 3.4. Correlations between exposure time and change in serum 25(OH)D

Two Pearson correlations were computed for the intervention group only to explore whether there were significant associations between cumulative exposure time and serum 25(OH)D in Week 4 and Week 8, respectively. The relation between cumulative exposure time until Week 4 and change in serum 25(OH)D from baseline to Week 4 showed a moderate correlation ($r$ = 0.47) that failed to reach statistical significance ($p$ = 0.09). The correlation between cumulative exposure time until Week 8 and change in serum 25(OH)D from baseline to Week 8 was low, and nonsignificant ($r$ = 0.10; $p$ = 0.75). This suggests that participants who spent more time under the lamp at the end of the study period did not achieve higher serum 25(OH)D levels compared to participants with a lower exposure time. As a comparison, for the control group the cumulative number of hours spent working on a desk (without a lamp) were correlated with the change in serum 25(OH)D. For both baseline to Week 4 and baseline to Week 8 the correlation was not statistically significant (respectively $r$ = -0.34; $p$ = 0.24; $r$ = -0.22; $p$ = 0.46). See Fig 4 for a visual overview of these analyses.

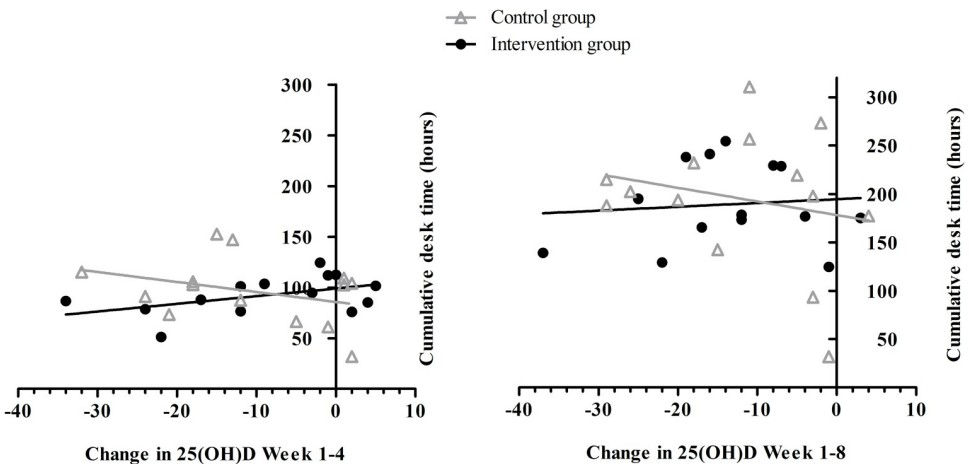

**Fig 4. Correlations between cumulative desk time and change in 25(OH)D from baseline to Week 4 and baseline to Week 8 for both groups.**

## 3.5. Correlations between change in serum 25(OH)D and change in sleep duration, sleep quality, midsleep timing and subjective fatigue

Pearson correlations on the data across all participants were investigated between change in serum 25(OH)D (baseline to Week 8) and change in respectively subjective measures of sleep duration, sleep quality, midsleep timing and fatigue. Table 7 shows that none of the change scores of the subjective indicators significantly correlated with change in serum 25(OH)D. For visual inspection purposes, we also conducted exploratory linear mixed model analyses for the intervention and control group separately to investigate associations between serum 25(OH)D and each of the subjective indicators. The output of these analyses can be viewed in the S1–S4 Tables and visualizations are shown in S1 Fig. These analyses only revealed a significant negative relationship between serum 25(OH)D and midsleep timing in the control group (see S4 Table).

## 4. Discussion

This study investigated whether using a low and safe dose of artificial UVB exposure applied in an office lighting solution could significantly attenuate the decline in serum 25(OH)D normally seen during the winter period in people living on the northern hemisphere. Our secondary aim was to explore whether changes in serum 25(OH)D correlated with changes in subjective sleep duration, midsleep timing, sleep quality, and general feelings of fatigue. The findings revealed that the current combination of received UVB dose and exposed BSA (~10%) in the intervention group was insufficient to attenuate a decline in serum 25(OH)D during the 8-week study period. In addition, in the current sample, there were no indications that declines in serum 25(OH)D during the winter period significantly correlated with changes in sleep duration, midsleep timing, sleep quality, and subjective fatigue.

**Table 7. Pearson correlations between change scores in serum 25(OH)D and sleep duration, midsleep, sleep quality and subjective fatigue.**

|  | Sleep duration | Midsleep | Sleep quality | Subjective Fatigue (CIS) |
|---|---|---|---|---|
| Change in serum 25(OH)D from baseline to Week 8 | -0.15 ($p = 0.50$) | <0.01 ($p = 0.99$) | 0.16 ($p = 0.43$) | 0.01 ($p = 0.95$) |

With respect to the ineffectiveness of the intervention on changing vitamin D status, it is important to take into account the average dose received in the intervention group and compare this to previous studies using similar doses. The average weekly exposure time in the intervention group was almost 24 hours, while the maximum exposure time per week could have been approximately 42 hours. This means that the estimated average weekly UVB dose received in this sample was about 0.9 SED, and about 7 SED cumulatively. In most participants, this dose was only exposed to the face, neck, and hands (~10% BSA). A previous study investigating a BSA exposure of approximately 14% (face, hands and arms) revealed that a cumulative dose of 13 SED received during seven subsequent days was sufficient to reveal a significant increase in serum 25(OH)D [25]. Another study examined three groups of participants during a 16-week UVB exposure period in winter receiving respectively a cumulative UVB dose of 4, 8, or 16 SED, and compared these to a control group [24]. Their findings revealed that a cumulative dose of 8 SED was just sufficient to prevent the decline in serum 25(OH)D that was observed in the control group. However, there was no significant difference in serum 25(OH)D over time between the 4 SED group and the control group. It is important to note that the exposed BSA was much larger in this previous study (i.e., 88%). Both studies [24,25] included a similar number of participants in each study group as in the current study. Considering these previous findings, it is likely that the cumulative dose in combination with the relatively small exposed BSA was insufficient to create meaningful changes in serum 25(OH)D.

We also investigated correlations between cumulative exposure time and 25(OH)D levels at Week 4 and Week 8 to explore dose-dependency. While a trend towards a moderate correlation was found at Week 4, this correlation turned out to be very small at Week 8. Again, this indicates that the total dose was likely too low to realize significant changes in serum 25(OH)D, even in participants with a higher amount of exposure hours. In this study, there was too little variation in exposed BSA to make reliable correlations with changes in serum 25(OH)D levels.

One way to increase the cumulative dose is to increase the duration of study period. It should be noted that the current study started in the middle of winter, when average vitamin D levels were already relatively low compared to average levels just after summer [27,28]. It would be useful to gain insight in whether low daily UVB doses could be sufficient to prevent a sharp decline in serum 25(OH)D when it is implemented directly after summer and extended towards early spring. When applied from October through March, this would yield a cumulative dose of approximately 21 SED. Moreover, the total dose could be further enhanced by also using the lighting device during weekend days and by making it portable so that it can be used at different places in the home.

Another important aspect that differentiates this study from previous studies was that the UVB exposure occurred during participants' daily work routines. This made it hard to control the actual number of hours spent under the lamp, participants' distance from the lamp and their seating position. Although the hours spent at the desk (under the lamp) were reported at the end of each day, these self-reports may be inaccurate or biased. Using presence sensors in future studies with a similar set-up may provide a more accurate estimate of the amount of exposure hours. In addition, even though a seating position was marked on participants' desks, and they were instructed not to move the lamp, changes to these positions may have happened and could have impacted the total received dose. The maximum dose of 0.3 SED per day was determined based on UVB measurements using a static mannequin seated at exactly 80 cm distance from the lamp. In real-life conditions, participants likely moved to different positions depending on the task they were performing (i.e., calling, typing, writing, reading), which may decrease the exposed BSA. This variation may have further decreased the actual

received dose. Previous research revealed that the role of exposed BSA in vitamin D synthesis efficacy is especially relevant when being exposed to relatively low UVB doses [37]. At the same time, another study found that skin surface of the hands and face are most effective in synthesizing vitamin D compared to other body sites [38]. Therefore, positioning of the device and the user are important aspects to consider in future research in order to maximize the exposed skin surface area of the face and hands.

The current findings did not reveal any significant relationships between change in serum 25(OH)D and change respectively sleep duration, midsleep timing, sleep quality, and fatigue. Likely, the current sample size was too small to reveal small to medium relationships between these variables. Previous research using larger sample sizes revealed significant associations between serum 25(OH)D and sleep indicators, usually indicating that higher vitamin D levels were related to better sleep scores, but the reverse has also been reported in some studies [12,39]. With respect to fatigue, intervention studies using oral supplements revealed that chronic fatigue symptoms were significantly reduced after correcting a vitamin D deficiency in people with chronic illness [40] as well as healthy people [41]. Future studies administrating a higher UVB dose, which prevents the decline in serum 25(OH)D in winter, should further investigate whether this may be beneficial for sleep and daily fatigue levels.

A limitation of the current study is that the control group did not receive a placebo intervention, which may have influenced participant behaviors in both groups differently. The pragmatic reason for this was the limited availability of lamps. However, it is unlikely that this has biased our main study objective (i.e., effects of receiving extra UVB on serum 25(OH)D in winter). We tried to control for (unconscious) changes in relevant behavioral patterns that could have impacted serum 25(OH)D. First, participants in both groups were instructed not to start using supplements during the study period. In addition, as revealed in previous studies, vitamin D related food intake has very little impact on serum 25(OH)D [10,11] and did not significantly differ between the two groups according to the questionnaires. Last, we monitored self-reported daily natural daylight exposure and investigated its effects on serum 25(OH)D. As expected [9], cumulative time spent outside in daylight during the winter period did not significantly correlate with changes in serum 25(OH)D. For self-report measures the risk of placebo effects is higher, therefore we only investigated correlations between (changes in) serum 25(OH)D and (changes in) sleep and fatigue over time across all study participants instead of potential effects of the intervention on these indicators. For further investigation of subjective health and wellbeing indicators in future research, we strongly recommend blinding the UVB manipulation such that participants in both conditions receive a lighting device with the same (visible) light spectrum.

A second limitation of this study was that we could not randomly allocate all participants to the two study conditions due to practical limitations (i.e., insufficient space for the lamp, no grounded electricity outlets, or changes in working conditions). This may have caused selection bias and it cannot be ruled out that this might have influenced our findings. For future studies, it is recommended to formulate stricter selection criteria so that all participants can be randomized.

## 5. Conclusion

The current findings revealed that a maximum low-dose UVB exposure of about 0.3 SED/day with an average weekly exposure time of 24 hours and a BSA of about 10% is insufficient to effectively attenuate decreases in serum 25(OH)D over a period of eight weeks in winter. Future research could explore whether a longer lasting exposure period starting directly after summer until the beginning of spring yields more promising results for improving vitamin D

status. In addition, future studies should focus on using different exposure positions of the device, maximizing the exposed skin surface of the hands, neck, and face. Continuation of the development and investigation of these home-based UVB lighting solutions could offer an accessible, safe, and easy-to-use method to maintain healthy vitamin D levels year-round, and potentially promote health and wellbeing in the general population.

## Supporting information

**S1 Checklist. CONSORT 2010 checklist of information to include when reporting a randomised trial\*.**
(DOC)

**S1 Fig. Associations between serum 25(OH)D and subjective measures of sleep quality, general feelings of fatigue, sleep duration, and midsleep timing in the intervention and control group.**
(TIF)

**S1 Table. Parameter estimates linear mixed model analysis; relationship between serum 25 (OH)D and sleep quality in the control group and intervention group.**
(PDF)

**S2 Table. Parameter estimates linear mixed model analysis; relationship between serum 25 (OH)D and sleep duration in the control group and intervention group.**
(PDF)

**S3 Table. Parameter estimates linear mixed model analysis; relationship between serum 25 (OH)D and general fatigue in the control group and intervention group.**
(PDF)

**S4 Table. Parameter estimates linear mixed model analysis; relationship between serum 25 (OH)D and midsleep in the control group and intervention group.**
(PDF)

**S1 File.**
(PDF)

## Author Contributions

**Conceptualization:** Laura M. Huiberts, Karin C. H. J. Smolders, Bianca M. I. van der Zande, Rémy C. Broersma, Yvonne A. W. de Kort.

**Data curation:** Laura M. Huiberts.

**Formal analysis:** Laura M. Huiberts.

**Funding acquisition:** Bianca M. I. van der Zande.

**Investigation:** Laura M. Huiberts.

**Methodology:** Laura M. Huiberts, Karin C. H. J. Smolders, Bianca M. I. van der Zande, Rémy C. Broersma.

**Project administration:** Laura M. Huiberts.

**Resources:** Bianca M. I. van der Zande, Rémy C. Broersma.

**Software:** Bianca M. I. van der Zande, Rémy C. Broersma.

**Supervision:** Karin C. H. J. Smolders, Yvonne A. W. de Kort.

**Visualization:** Laura M. Huiberts, Karin C. H. J. Smolders.

**Writing – original draft:** Laura M. Huiberts.

**Writing – review & editing:** Laura M. Huiberts, Karin C. H. J. Smolders, Bianca M. I. van der Zande, Rémy C. Broersma, Yvonne A. W. de Kort.

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
