## [Decision Letter · Decision Letter 0]

29 Mar 2022

PONE-D-21-33569Using a low-dose UVB lighting solution during working hours: An explorative investigation towards the effectivity in maintaining healthy vitamin D levelsPLOS ONE

Dear Dr. Huiberts,

Thank you for submitting your manuscript to PLOS ONE. After careful consideration, we feel that it has merit but does not fully meet PLOS ONE’s publication criteria as it currently stands. Therefore, we invite you to submit a revised version of the manuscript that addresses the points raised during the review process.

We look forward to receiving your revised manuscript.

Kind regards,

Walid Kamal Abdelbasset, Ph.D.

Academic Editor

PLOS ONE

Journal Requirements:

2. Thank you for stating the following in the Acknowledgments/ Funding Section of your manuscript: 

This research was funded by Signify, Eindhoven, the Netherlands.

This research was funded by Signify, Eindhoven, the Netherlands. Funders had a role in the study design, data collection (only light measurements), and preparation of the manuscript. Funders had no role in data collection from participants, data analysis, and decision to publish. 

Reviewers' comments:

Reviewer's Responses to Questions

**Comments to the Author**

1. Is the manuscript technically sound, and do the data support the conclusions?

Reviewer #1: Yes

Reviewer #2: Yes

Reviewer #3: Partly

2. Has the statistical analysis been performed appropriately and rigorously? 

Reviewer #1: Yes

Reviewer #2: Yes

Reviewer #3: Yes

3. Have the authors made all data underlying the findings in their manuscript fully available?

Reviewer #1: Yes

Reviewer #2: No

Reviewer #3: Yes

4. Is the manuscript presented in an intelligible fashion and written in standard English?

Reviewer #1: Yes

Reviewer #2: Yes

Reviewer #3: Yes

5. Review Comments to the Author

Reviewer #1: This study examined whether daily low-dose UVB exposure could maintain healthy serum 25(OH)D during winter. The findings did not reveal a statistically significant effect on serum 25(OH)D.

The work is thorough and very well written.

I think it can be accepted for publication as is.

Reviewer #2: Overall, I found this study as a sound and well written manuscript , however , I am not sure if it is adding much to the current knowledge. Also, the experiments design could have be improved by adding different UV light types/doses/durations.

Reviewer #3: Dear authors thanks for submitting this work to this journal but please make the attached required correction for your paper to be sound for publication on this journal and follow the required instructions

6. PLOS authors have the option to publish the peer review history of their article (what does this mean?). If published, this will include your full peer review and any attached files.

Reviewer #1: **Yes: **Marilda Aparecida Milanez Morgado de Abreu

Reviewer #2: No

Reviewer #3: **Yes: **Marwa Eid

---

## [Author Response · Author response to Decision Letter 0]

20 Apr 2022

We would like to thank the reviewers for their feedback on our manuscript. In the revised manuscript we have addressed their comments and marked all changes. Below, we have provided a point-by-point response to the reviewers’ comments. We hope that the changes in the manuscript address the points that were raised to the reviewers’ satisfaction.

Reviewer #2:

The reviewer wrote: “Overall, I found this study as a sound and well written manuscript, however, I am not sure if it is adding much to the current knowledge. Also, the experiments design could have been improved by adding different UV light types/doses/durations.”

We would like to thank the reviewer for his/her comment. To our knowledge, this is the first study which investigates the effectiveness of a daily low-dose and low body surface area UVB exposure intervention in the home/office setting to sustain healthy serum 25(OH)D during the winter period. As explained in the introduction, we think that this could offer an unobtrusive, accessible, and safe solution in the future to improve serum 25(OH)D year-round with potential secondary health benefits, also compared to oral supplementation only [1]. We feel that the recommendations in the discussion section (e.g., regarding dose, exposed body surface area, and duration of use) are especially relevant for further development and testing of these types of devices for use in the home and/or office setting. We agree that, based on the current findings, new (types of) lamps should be developed and investigated in the future that can improve effectiveness for sustaining healthy serum 25(OH)D in winter. We addressed this in the discussion section, lines 442-469 in ‘Revised Manuscript with Track Changes’.

1. Tuckey RC, Cheng CYS, Slominski AT. The serum vitamin D metabolome: what we know and what is still to discover. J Steroid Biochem Mol Biol. Elsevier; 2018;186:4–21.

Reviewer #3:

The reviewer wrote: “Dear authors thanks for submitting this work to this journal but please make the attached required correction for your paper to be sound for publication on this journal and follow the required instructions”

We thank the reviewer for the feedback on our manuscript. A first important remark concerned the necessity for the control group to receive a placebo treatment. We agree with the reviewer that this would be preferable, but due to pragmatic reasons (the availability of lamps) this was not possible for the current study. However, we would like to point out that for our main study objective (effects of receiving extra UVB exposure in winter on serum 25(OH)D), the chance of a placebo effect is highly unlikely as vitamin D synthesis in the skin is purely photon-driven. Moreover, we checked and controlled for potential behavioral changes in either of the groups that may have affected serum 25(OH)D (i.e., vitamin D related food intake and time spent outside in daylight). Even though during the study period (winter) at our latitude it was not expected that extra time spent outside would affect serum 25(OH)D [2], we still included cumulative time spent outside in daylight as a control variable in our analysis. This confirmed our expectation, as it did not significantly affect 25(OH)D levels between January and March in our study sample. Furthermore, vitamin D related food intake was not significantly different between the two groups, and the contribution of vitamin D intake via only food intake on serum 25(OH)D is very limited [3, 4]. In addition, we instructed all participants at the beginning and during the course of the study not to start taking vitamin D supplements. We further elaborated on these points in the discussion section (line 481 – 492 in ‘Revised Manuscript with Track Changes’). One participant in the control group was taking a daily multivitamin which contained a low vitamin D dose (the participant found this out after the study had already started). Therefore, we decided to keep this participant in the study and perform the main analysis with and without this participant. The results were not affected by this (see lines 341-342 in ‘Revised Manuscript with Track Changes’). For the subjective indicators (i.e., sleep and fatigue), on the other hand, a placebo treatment (lamp with the same visible light spectrum without additional UVB) would be necessary for testing effects in the future (addressed in the discussion section, line 492-497 in ‘Revised Manuscript with Track Changes’). Therefore, we only investigated relationships between changes in serum 25(OH)D and subjective indicators over time across all study participants, instead of analyzing the effect of the intervention on these subjective indicators.

2. Webb AR. Who, what, where and when—influences on cutaneous vitamin D synthesis. Prog Biophys Mol Biol. 2006;92(1):17–25.

3. Lamberg-Allardt C. Vitamin D in foods and as supplements. Prog Biophys Mol Biol. 2006;92(1):33–8.

4. Troesch B, Hoeft B, McBurney M, Eggersdorfer M, Weber P. Dietary surveys indicate vitamin intakes below recommendations are common in representative Western countries. Br J Nutr. Cambridge University Press; 2012;108(4):692–8.

In the method section, the reviewer wrote: “How was sample size determined? add the calculation method” and “What was the method used to generate the random allocation sequence. Explain the type of randomization”

We rearranged section 2.2. and added the necessary information. We addressed the sample size calculation program, input and output here. The full data, visualizations and explanation regarding the calculation method are available in the study protocol. The link to this document was also added in this section. Moreover, we added information about the generation of the random allocation sequence.

The reviewer also wrote: “Intervention at home may affect the results as participants may not follow the instructions perfectly”

We agree that adherence to the study protocol could not be controlled entirely, this limitation is inherent to field studies. We did, however, ask participants to complete a daily diary in which they reported the number of hours worked in their home office (where the lamp was placed, and automatically switched on between 8:50 and 17:10). Response rates on this daily diary were very high (>98%; see results section). Nevertheless, we addressed this potential limitation further in the discussion section (line 451-457), suggesting a point for improvement in future studies with a similar set-up.

We adjusted each of the other points suggested by the reviewer, these changes can be found in the file 'Revised Manuscript with Track Changes’.

---

## [Decision Letter · Decision Letter 1]

19 May 2022

PONE-D-21-33569R1Using a low-dose UVB lighting solution during working hours: An explorative investigation towards the effectivity in maintaining healthy vitamin D levelsPLOS ONE

Dear Dr. Huiberts,

Thank you for submitting your manuscript to PLOS ONE. After careful consideration, we feel that it has merit but does not fully meet PLOS ONE’s publication criteria as it currently stands. Therefore, we invite you to submit a revised version of the manuscript that addresses the points raised during the review process.

We look forward to receiving your revised manuscript.

Kind regards,

Walid Kamal Abdelbasset, Ph.D.

Academic Editor

PLOS ONE

Journal Requirements:

Reviewers' comments:

Reviewer's Responses to Questions

**Comments to the Author**

1. If the authors have adequately addressed your comments raised in a previous round of review and you feel that this manuscript is now acceptable for publication, you may indicate that here to bypass the “Comments to the Author” section, enter your conflict of interest statement in the “Confidential to Editor” section, and submit your "Accept" recommendation.

Reviewer #1: All comments have been addressed

Reviewer #3: (No Response)

Reviewer #4: (No Response)

2. Is the manuscript technically sound, and do the data support the conclusions?

Reviewer #1: Yes

Reviewer #3: Yes

Reviewer #4: Yes

3. Has the statistical analysis been performed appropriately and rigorously? 

Reviewer #1: Yes

Reviewer #3: Yes

Reviewer #4: Yes

4. Have the authors made all data underlying the findings in their manuscript fully available?

Reviewer #1: Yes

Reviewer #3: Yes

Reviewer #4: No

5. Is the manuscript presented in an intelligible fashion and written in standard English?

Reviewer #1: Yes

Reviewer #3: Yes

Reviewer #4: Yes

6. Review Comments to the Author

Reviewer #1: The authors followed the reviewers' suggestions and it is better. I think it can be accepted for publication as is.

Reviewer #3: the required modification attached in a word file kindly do the required modifications for the soundness of your paper to be puplished

Reviewer #4: In general, the manuscript is well-written, and presents interesting results from a pilot study of remediation or prevention of vitamin D deficiency in at-home workers.

In the limitations discussion, please include some discussion of the issues around inability to perform randomization for all subjects. While it is reasonable that there are few issues with this, it is up to the authors to make the case --- and, it cannot be said definitively that this did not play a role in the results.

In Lines 235-236 the authors describe a slightly troubling methodology where data are thrown out if they are "outliers". The first part of the results section should describe all such data quality issues. How many data were discarded? Could these have important effects on the conclusions?

In general, it is very poor practice to discard data and then present results based on the "cleaned" data alone. It is better practice to present the analysis on the actual data, then present an alternative analysis on the "cleaned" data (cf. per protocol versus intent-to-treat).

Rather than providing demographics in the Methods section, provide a standard "Table 1" in the Results section that gives demographic and baseline characteristics for the study population by arm and overall.

For section 3.4 and 3.5, please provide plots of the data.

Also, provide the same analysis for the control group as was provided for the intervention group. Otherwise, it makes it seem as though the authors do not wish to make the comparison with the control group.

In the Study Protocol, the authors plan to use linear mixed models to perform these exploratory analyses. This pre-planned analysis should be done and reported, if only briefly. If it is too lengthy it can be included as a Supplementary analysis. Figures analogous to Figure 2 should also be created for these analyses.

In the Conclusions please provide a quantitative estimate of the actual benefit found, along with confidence limits. It is fine that it is not statistically significant. It is also quite useful to other researchers to be able to evaluate the likely magnitude of any effects.

Generally, for linear mixed model analysis as in any linear model analysis, the authors should provide a brief evaluation of the aptness of the modeling. This is typically done by evaluating the residuals from the model for symmetry, independence, homoscedasticity, and normality (of less importance arguably).

Note that the authors' evaluation of normality initially is a common misguided activity --- the residuals from linear model analysis are typically assumed to be normally distributed, not the actual data. Considering the structure of the authors' data will immediately make it clear why the original data do not need to be normally distributed, if the underlying linear model is correct.

The following are minor comments:

Line 112: Replace "revealed" with "required".

Line 239: Do not capitalize "Linear Mixed Model".

Line 240: Put a comma after "random intercept". The sentence is wrong otherwise.

Line 281: Do the authors mean "upper arms" here? It seems unlikely that under-arms were exposed by rolling up sleeves.

Table 4: What is the "B estimate"? Since this is derived from a linear mixed model, the authors should also provide the estimates for the random effects.

7. PLOS authors have the option to publish the peer review history of their article (what does this mean?). If published, this will include your full peer review and any attached files.

Reviewer #1: **Yes: **Marilda Aparecida Milanez Morgado de Abreu

Reviewer #3: No

Reviewer #4: No

---

## [Author Response · Author response to Decision Letter 1]

2 Jul 2022

We thank the reviewers for the second round of feedback on our manuscript. In the revised manuscript we have addressed the comments and marked all changes. Below, we have provided a point-by-point response to the reviewers’ comments. We hope that the changes in the manuscript address the points that were raised to the reviewers’ satisfaction.

Reviewer #3:

The reviewer wrote:

“Sample size calculation revealed 37 participants so you should enroll equal or more than to get sound results and allow generalization so you should justify this”

We would like to point out that the sample size calculation resulted in 34 participants (not 37, see study protocol page 16). During the participant recruitment phase, 35 participants were recruited. From these, 29 returned the signed informed consent form (see Figure 1 of the manuscript). Due to the predetermined study period timing which should be in winter (i.e., January to March for all participants) and practical limitations (i.e., only 14 UVB-containing lamps were manufactured), it was decided to start the study with 29 participants who gave written consent. We further explained these practical considerations in the revised manuscript (see lines 115-121 in ‘Revised Manuscript with Track Changes’). In addition, the potential main reasons for not finding any significant effects of the intervention in the current study were further explained in the discussion section (i.e., half of the expected exposure time and a small body surface area exposure due to angle of the light exposure). Three extra participants in each group would yield a detection of only a slightly smaller effect in the current study (effect size d of 1.05 instead of 1.14, assuming a power of .90 and 17 participants per group). The actual effect to be detected in the current study – if there was any – was substantially smaller than expected due to less than expected UVB exposure.

The reviewer wrote:

“These parameters not clear and what SD mean is it standard deviation and why you added it?”

mWA/m2 is the measurement unit of Actinic weighted UVB irradiance (the sub A stands for Actinic). The CIE effective UV threshold in commercial lamps should be limited to < 30 JA/m2 per 8 hours within a time span of 24 hours, which results in an upper limit for actinic UV irradiance of less than 1 mWA/m2, which was the case for the devices we used (i.e., ~ 0.76 mWA/m2 ). This was also explained at the beginning of this section (lines 160-167 in the revised manuscript) with reference to reference number 26 (where this is further explained). SD indeed stands for standard deviation, we added these SD’s because there was a slight variation in actinic UV weighted UVB irradiance between the 14 lamps used.

 

Reviewer #4

The reviewer wrote:

“In the limitations discussion, please include some discussion of the issues around inability to perform randomization for all subjects. While it is reasonable that there are few issues with this, it is up to the authors to make the case --- and, it cannot be said definitively that this did not play a role in the results.”

We thank the reviewer for this suggestion and agree that randomization for all subjects would have been preferred. We expanded on this limitation to the discussion section (see lines 546-550 in ‘Revised Manuscript with Track Changes’).

The reviewer wrote:

“In Lines 235-236 the authors describe a slightly troubling methodology where data are thrown out if they are "outliers". The first part of the results section should describe all such data quality issues. How many data were discarded? Could these have important effects on the conclusions?”

and

“In general, it is very poor practice to discard data and then present results based on the "cleaned" data alone. It is better practice to present the analysis on the actual data, then present an alternative analysis on the "cleaned" data (cf. per protocol versus intent-to-treat).”

We would like to clarify that in the current analyses, we included all data, except for the explorative analyses on midsleep timing and sleep duration (section 3.5.), for which there were three illogical values (due to participant reporting) which could not be translated into a time-value (see also Table 2 in the results section, where the N was given added along with the mean values where this was deviant from 14). There were no outliers > 3 SD’s in the variables used for the main analyses, so no observations were removed. We clarified this in the beginning of the results section (lines 297-301 in ‘Revised Manuscript with Track Changes’).

The reviewer wrote:

“Rather than providing demographics in the Methods section, provide a standard "Table 1" in the Results section that gives demographic and baseline characteristics for the study population by arm and overall.”

We added this information at the beginning of the Results section (see Table 2), instead of in the Methods section.

 

The reviewer wrote: 

“For section 3.4 and 3.5, please provide plots of the data.

Also, provide the same analysis for the control group as was provided for the intervention group. Otherwise, it makes it seem as though the authors do not wish to make the comparison with the control group. In the Study Protocol, the authors plan to use linear mixed models to perform these exploratory analyses. This pre-planned analysis should be done and reported, if only briefly. If it is too lengthy it can be included as a Supplementary analysis. Figures analogous to Figure 2 should also be created for these analyses.”

We added plots of the analyses of section 3.4 in the Results section (see Figure 4). We also ran the correlational analyses in section 3.4 for the control group. However, it should be noted that there is no real cumulative lamp exposure in the control group as this group did not receive a lamp at their work desk. Thus, we correlated cumulative desk time to change in serum 25(OH)D from baseline to Week 4 and baseline to Week 8, but only in the intervention group this was equivalent to actual cumulative UVB exposure.

With regard to section 3.5 it should be noted that it was never our intention to make direct comparisons between the intervention and the control group for the subjective outcome measures due to the high chance of a placebo effect because study conditions were not blinded. Participants in the intervention group knew that they were exposed to additional UVB as they received a lamp, while participants in the control group did not receive a lamp and knew they were not exposed to additional UVB. Nevertheless, for visual inspection purposes we added plots of the associations between 25(OH)D and the subjective outcome measures for both conditions together with the output of the exploratory linear mixed model analyses as described in the study protocol (see Supporting Information Tables S1 – S4, and Figure S1; and lines 439 – 445 in ‘Revised Manuscript with Track Changes’). 

The reviewer wrote:

“In the Conclusions please provide a quantitative estimate of the actual benefit found, along with confidence limits. It is fine that it is not statistically significant. It is also quite useful to other researchers to be able to evaluate the likely magnitude of any effects.”

We agree with the reviewer that it is important to give some quantitative estimates of the benefits found. However, we would like to refrain from reporting statistics in the Discussion/Conclusion section as this is not common practice. However, we added extra quantitative estimates in the Results section to quantify the actual benefit. First, we decided to conduct our main analyses according to the ANCOVA as specified in the study protocol (see page 35), estimating the effect of the intervention on change in serum 25(OH)D from baseline to Week 8. This analysis yields a partial eta squared effect size together with the coefficient of the effect of the intervention (see Table 5 and lines 336 – 341 in ‘Revised Manuscript with Track Changes’). In addition, we added the actual difference scores between conditions (controlled for baseline scores and time spent outside) at Week 4 and Week 8, including the 95% confidence intervals, based on the subsequent linear mixed model analysis. We also improved the figure belonging to this analysis (Figure 2) by only showing Week 4 and 8 and not baseline values. The baseline values were estimated based on the raw values, and only the values at Week 4 and 8 resulted from the LMM analysis. Last, to visualize changes over time including baseline values across all participants, we conducted a separate LMM analysis including measurement (baseline, Week 4, Week 8) to obtain all values from one model for the figure (see Figure 3). 

The reviewer wrote:

“Generally, for linear mixed model analysis as in any linear model analysis, the authors should provide a brief evaluation of the aptness of the modeling. This is typically done by evaluating the residuals from the model for symmetry, independence, homoscedasticity, and normality (of less importance arguably).

Note that the authors' evaluation of normality initially is a common misguided activity --- the residuals from linear model analysis are typically assumed to be normally distributed, not the actual data. Considering the structure of the authors' data will immediately make it clear why the original data do not need to be normally distributed, if the underlying linear model is correct.”

We agree with the reviewer that normality of the residuals (assumption for LMM analysis) is important to inspect instead of (primarily) normality of the raw data. We checked the symmetry, independence, homoscedasticity, and normality of the residuals, and verified the aptness of the modeling in the statistical analyses section (see lines 271-286 in ‘Revised Manuscript with Track Changes’).

The reviewer wrote:

“The following are minor comments:

Line 112: Replace "revealed" with "required".

Line 239: Do not capitalize "Linear Mixed Model".

Line 240: Put a comma after "random intercept". The sentence is wrong otherwise.

Line 281: Do the authors mean "upper arms" here? It seems unlikely that under-arms were exposed by rolling up sleeves.” 

We corrected / clarified the points above in the revised manuscript.

The reviewer wrote:

“Table 4: What is the "B estimate"? Since this is derived from a linear mixed model, the authors should also provide the estimates for the random effects.” 

The B estimate label was changed to “Estimate (unstandardized coefficient)” (derived from the linear mixed model analysis). Random effects estimates were added to this table (Table 6 in the revised manuscript).

---

## [Decision Letter · Decision Letter 2]

6 Mar 2023

Using a low-dose ultraviolet-B lighting solution during working hours: An explorative investigation towards the effectivity in maintaining healthy vitamin D levels

PONE-D-21-33569R2

Dear Dr. Huiberts,

We’re pleased to inform you that your manuscript has been judged scientifically suitable for publication and will be formally accepted for publication once it meets all outstanding technical requirements.

Kind regards,

Walid Kamal Abdelbasset, Ph.D.

Academic Editor

PLOS ONE

Additional Editor Comments (optional):

Reviewers' comments:

Reviewer's Responses to Questions

**Comments to the Author**

1. If the authors have adequately addressed your comments raised in a previous round of review and you feel that this manuscript is now acceptable for publication, you may indicate that here to bypass the “Comments to the Author” section, enter your conflict of interest statement in the “Confidential to Editor” section, and submit your "Accept" recommendation.

Reviewer #1: All comments have been addressed

Reviewer #3: All comments have been addressed

Reviewer #4: All comments have been addressed

2. Is the manuscript technically sound, and do the data support the conclusions?

Reviewer #1: Yes

Reviewer #3: Yes

Reviewer #4: (No Response)

3. Has the statistical analysis been performed appropriately and rigorously? 

Reviewer #1: Yes

Reviewer #3: Yes

Reviewer #4: (No Response)

4. Have the authors made all data underlying the findings in their manuscript fully available?

Reviewer #1: Yes

Reviewer #3: Yes

Reviewer #4: (No Response)

5. Is the manuscript presented in an intelligible fashion and written in standard English?

Reviewer #1: Yes

Reviewer #3: Yes

Reviewer #4: (No Response)

6. Review Comments to the Author

Reviewer #1: The manuscript is well-written, and presents interesting results from a study of prevention of vitamin D deficiency in at-home workers. I think that manuscript is adequate for publication.

Reviewer #3: All the required modifications had been done and all questions had been answered so it seems to be sound for publication

Reviewer #4: The authors have done a good job in responding to the criticism. The supplementary figure is very nicely done, also.

7. PLOS authors have the option to publish the peer review history of their article (what does this mean?). If published, this will include your full peer review and any attached files.

Reviewer #1: **Yes: **Marilda AM Morgado de Abreu

Reviewer #3: No

Reviewer #4: No

---

## [Editor Report · Acceptance letter]

22 Mar 2023

PONE-D-21-33569R2 

Using a low-dose ultraviolet-B lighting solution during working hours: An explorative investigation towards the effectivity in maintaining healthy vitamin D levels 

Dear Dr. Huiberts:

I'm pleased to inform you that your manuscript has been deemed suitable for publication in PLOS ONE. Congratulations! Your manuscript is now with our production department. 

Kind regards, 

on behalf of

Dr. Walid Kamal Abdelbasset 

%CORR_ED_EDITOR_ROLE%

PLOS ONE